# Rapid inverse design of metamaterials based on prescribed mechanical behavior through machine learning

Chan Soo Ha [1,8], Desheng Yao [2,3,8], Zhenpeng Xu[2,3], Chenang Liu[4], Han Liu [5], Daniel Elkins[1,6], Matthew Kile[1], Vikram Deshpande [7] ✉, Zhenyu Kong [6] ✉, Mathieu Bauchy[3] ✉ & Xiaoyu (Rayne) Zheng [1,2,3] ✉

Designing and printing metamaterials with customizable architectures enables the realization of unprecedented mechanical behaviors that transcend those of their constituent materials. These behaviors are recorded in the form of response curves, with stress-strain curves describing their quasi-static footprint. However, existing inverse design approaches are yet matured to capture the full desired behaviors due to challenges stemmed from multiple design objectives, nonlinear behavior, and process-dependent manufacturing errors. Here, we report a rapid inverse design methodology, leveraging generative machine learning and desktop additive manufacturing, which enables the creation of nearly all possible uniaxial compressive stress–strain curve cases while accounting for process-dependent errors from printing. Results show that mechanical behavior with full tailorability can be achieved with nearly 90% fidelity between target and experimentally measured results. Our approach represents a starting point to inverse design materials that meet prescribed yet complex behaviors and potentially bypasses iterative design-manufacturing cycles.

The intrinsic mechanical behavior of bulk materials (e.g., metals, ceramics, polymers) can be experimentally characterized by the application of force and the measurement of the resulting deformation, yielding stress–strain curves. For example, under tensile loading, the mechanical behavior of brittle materials such as ceramics is characterized by a stress–strain curve with a linear region followed by a sharp termination; elastomers display superelasticity, characterized by a rapidly rising concave-up stress–strain curve without a noticeable linear region. For homogenous materials such as metals, ceramics, and polymers, responses to loading are dictated by intrinsic microstructure, such as crystal structure, atomic bonding, and the size and mass of the constituent molecules/atoms, in addition to the presence

of stochastic microscopic defects. As a result, there is little room to tailor these materials' responses to loads besides altering the intrinsic microstructure of the base materials.

Additive manufacturing (AM) allows mechanical properties to be tailored in ways that are impossible in bulk materials, via the microarchitecture design of three-dimensional (3D) metamaterials. These materials can exhibit unusual properties such as negative Poisson's ratio[1–3], negative compressibility[4,5], ultralightness and ultrastiffness, shape recoverability[6–8], and multiple stable states[9–11]. These architected materials achieve previously unattainable region in the material selection chart (e.g., so-called Ashby charts of density vs. Young's modulus or strength)[12,13]. Architected materials manifesting these

[1]Department of Mechanical Engineering, Virginia Tech, Blacksburg, VA, USA. [2]Department of Material Science and Engineering, University of California, Berkeley, CA, USA. [3]Department of Civil and Environmental Engineering, University of California, Los Angeles, CA, USA. [4]Industrial Engineering and Management, Oklahoma State University, Stillwater, OK, USA. [5]Department of Computer Science and Technology, Sichuan University, Chengdu, China. [6]Grado Department of Industrial and Systems Engineering, Virginia Tech, Blacksburg, VA, USA. [7]Department of Engineering, University of Cambridge, Cambridge, UK. [8]These authors contributed equally: Chan Soo Ha, Desheng Yao. ✉e-mail: vsd20@cam.ac.uk; zkong@vt.edu; bauchy@ucla.edu; rayne23@berkeley.edu

properties have been typically designed by forward design approaches, topology optimizations[14,15] and, more recently, machine learning (ML)[16–22]. The forward approaches iteratively adjust design parameters of architected materials (such as unit cell size, wall thickness, and so forth) until measured or simulated properties satisfy prescribed design criteria, which usually require substantial prior knowledge of experienced designers. While the topology optimizations and ML-based design approaches have shown the potential to yield designs that provide desired property values, they have yet to accurately capture all of the required mechanical behaviors in practice, due to non-unique response-to-design mapping and challenges in simultaneously representing a large number of variables. These design approaches are further complicated by the presence of manufacturing defects, process variabilities, and uncertainties[23,24], which necessitate substantial calibrations to account for defects in additively manufactured samples with hundreds to millions of spatial constituents as each member independently contributes to the realization of target responses[25,26]. As a result of these challenges, the actual mechanical properties of fabricated samples often substantially deviate from the designed properties[25,26], which, if not considered, could lead to suboptimal or catastrophic failure on application.

In this work, we present a rapid inverse design methodology that can produce designs that replicate tailored mechanical behaviors upon loading via ML and desktop 3D printing. Our approach leverages generative inverse and surrogate forward neural network (NN) models (details of the ML workflow shown in Fig. 1a), where the input comprises a user-defined uniaxial compressive stress–strain curve (in the form of curve features $\{X^T\}$) and fabrication parameters (the maximum build volume dimensions ($L^3$) and minimum printable feature size

($s_{min}$) of a given 3D printer). The output is a set of optimal design parameters $\{Y\}$ that describes a digital lattice design that, once 3D-printed and tested, will replicate the prescribed stress–strain curve (Fig. 1b–e). To achieve this, we develop a family of architectural unit cells capable of capturing distinct curve shapes under both monotonic and cyclic compressive loadings that cover a wide range of mechanical behaviors of a cellular solid. These cells serve as building blocks for creating training data sets differentiated by two distinct (brittle and flexible) polymeric base materials, from which our ML pipeline learns the relationship between various mechanical behaviors, topology, and process-dependent manufacturing errors, and generates printable lattice replicating the target stress–strain curve. We demonstrate the inverse design of arbitrary stress–strain curves which represent the entire mechanical responses under monotonic and cyclic loading. We also show experimentally that additional tailored mechanical behaviors can be realized by graphically modifying the local geometric features of a stress–strain curve. This methodology allows for the rapid creation of materials with fully tailorable mechanical behavior while accounting for manufacturing process errors and nonlinear behavior.

## Results

### Overview of the generative ML approach
We implemented a generative ML pipeline composed of inverse prediction and forward validation modules where each module is composed of five distinct NN models (Fig. 1a; details of the ML pipeline provided in Supplementary Note 1). Each NN model in the inverse prediction module predicts a set of design parameters $\{Y\}$ for a given target curve feature $\{X^T\}$, whereas the forward validation module outputs the predicted curve features $\{X^P\}$ for each set of the predicted

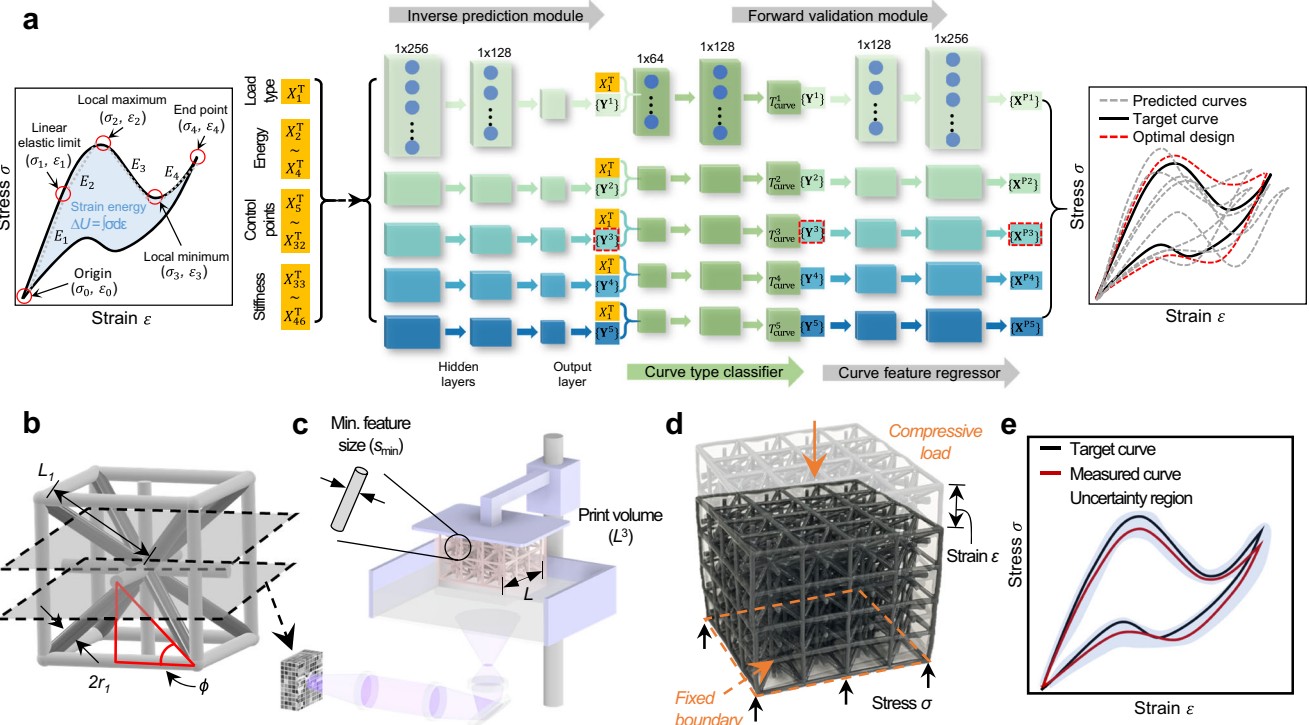

**Fig. 1 | Overview of the ML-based rapid inverse design methodology.**
**a** Schematic of the generative ML pipeline presented in this work. By taking the target uniaxial compressive behavior in the form of curve features $\{X^T\}$ as the input data, the inverse prediction module of our ML approach predicts five sets of design candidates $\{Y^k\}$ (described by the cell type ($T_{cell}$), characteristic angle ($\phi$) and radius-to-length ratio ($r_1/L_1$)), where $k$ ranges from 1 to 5. These design candidates are then passed to forward validation module to estimate the response $\{X^{Pk}\}$ of the design candidates. Each of these responses $\{X^{Pk}\}$ is compared to the target curve

feature $\{X^T\}$ for selection of the optimal design. **b** CAD model generated based on the optimal design selected in **a**. **c** A schematic of 3D printing system with specific fabrication parameters (minimal feature size $s_{min}$ and maximum printing volume $L^3$). **d** Printed sample based on the optimal design predicted by the generative ML pipeline under compressive loading. **e** A comparison between the target (black) and measured (red) compressive stress–strain curves. The uncertainty region (blue-shaded area) represents the process variability obtained through the testing of multiple samples.

design parameters and determines the optimal set via evaluating the differences between predicted and target curve features (details of optimal design selection process in Supplementary Note 1.2). This embedded approach solves the non-unique response-to-design mapping challenge in inverse design[27,28] (e.g., several micro-architectural features may give same output curves). During this process, the curve type classifier in the forward validation module estimates the type of predicted stress–strain curves using the design parameters predicted from the inverse design module. This curve type along with such predicted design parameters are fed into each NN model of the forward module for the prediction of stress–strain curve features $\{X^p\}$. Thereby, as the optimal design is chosen via a direct comparison of curve features, our approach ensures the uniqueness of the solution and hence bypasses the potential one-to-many mapping issue that can occur in the typical inverse design approach.

## Curve design space

The design region of our generative ML approach was formulated as a dimensionless design space enclosing the arbitrary mechanical behavior of cellular materials under both monotonic and cyclic uniaxial compression, where the $x$-axis specifies the strain $\varepsilon$ and the $y$-axis specifies the relative compressive strength normalized to the yield strength of a given polymeric base material $\sigma/\sigma_{ys}$ (the full curve design space is highlighted by a black dotted region in Fig. 2a). This dimensionless design space comprises a series of subdesign spaces classified by the elastic limit ($\varepsilon_{ys}$ or $\sigma_{ys}/E_s$) of each available polymeric base material (gray envelopes in Fig. 2a). This representation allows for the inclusion and visualization of nearly all possible stress–strain curve shapes depending on the choice of polymeric base materials. Envelopes of the subdesign space were specified by the theoretical upper bounds of the elastic stiffness[29] and the yield strength[30] of isotropic

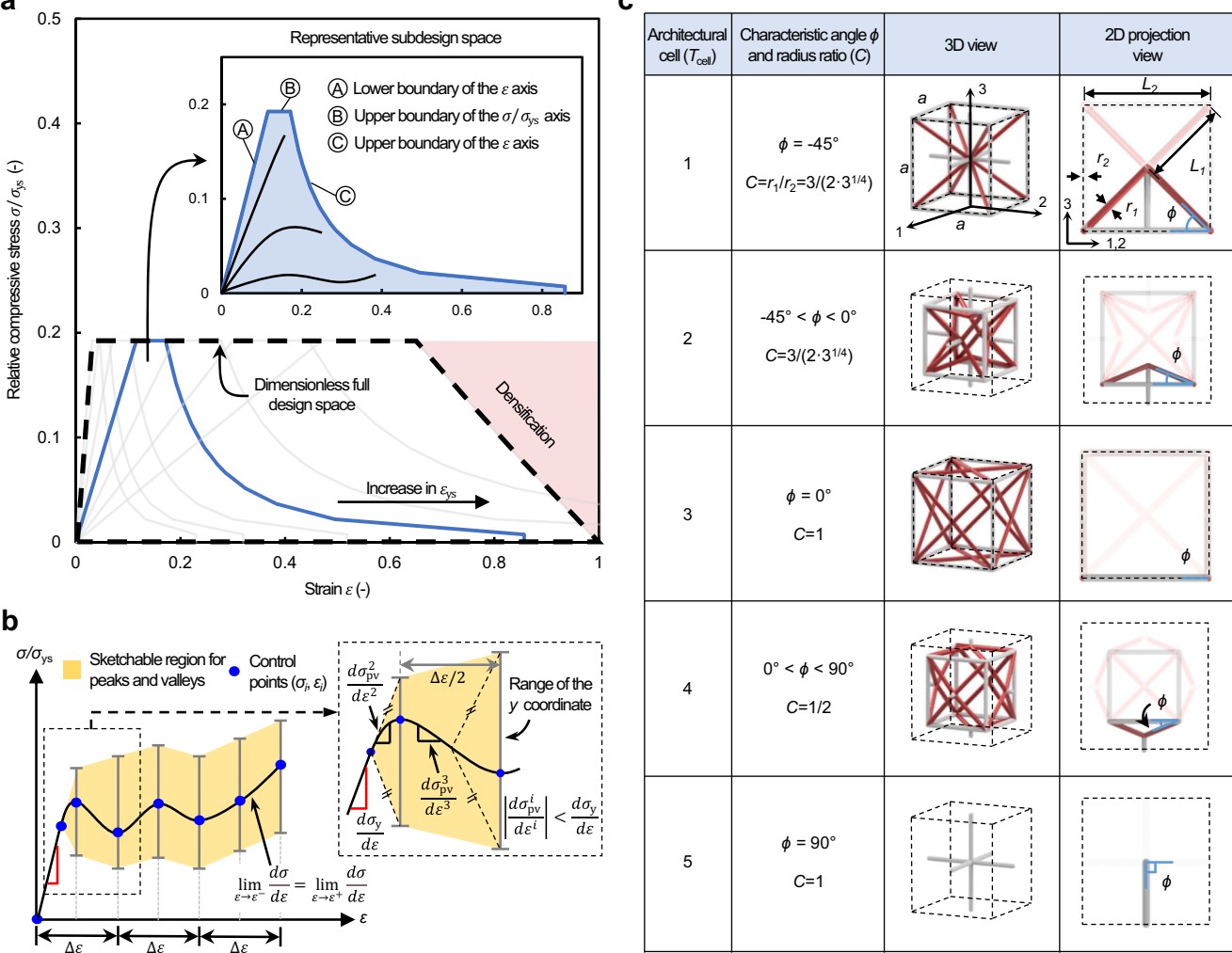

**Fig. 2 | Design space, plottable stress–strain curve paths and architectural cells. a** Full stress–strain curve design space composed of a series of subdesign spaces in a dimensionless plot, where the $x$ axis specifies the strain and the $y$ axis specifies the relative compressive strength $\sigma/\sigma_{ys}$ ($\sigma_{ys}$ denotes the yield strength of the base material). Each subdesign space is associated with a unique base material described by its elastic limit $\varepsilon_{ys}$ and constructed with three boundaries (Ⓐ, Ⓑ, and Ⓒ in inset). A representative subdesign space is shown as a blue envelop. Example stress–strain curves are also shown in the inset figure. **b** Design rules for plotting target stress–strain curves. A target curve, described by control points ($\varepsilon_i, \sigma_i$), starts with a straight line, followed by peaks and valleys. Error bars represent bounds of the peaks and valleys determined by the tangent modulus which is lower or equal to

the linear-elastic slope (i.e., $|(d\sigma_{pv}{}^i)/(d\varepsilon^i)| \leq (d\sigma_y{}^1)/(d\varepsilon^1)$, where $i = 2,...,\ \max(N_{pv})$ and $\max(N_{pv})$ denotes the maximum number of the peaks and valleys). **c** Architectural cells with cubic symmetry. A variation in the characteristic angle ($\phi$) from −45 to 90 degrees results in an architectural transformation from a compound truss comprising simple and body-centered cubic trusses ($T_{cell} = 1$), to an auxetic truss ($T_{cell} = 2$), to a reinforced face-centered truss ($T_{cell} = 3$), to a simple cubic truss combined with convex square pyramids truss ($T_{cell} = 4$), and to a simple cubic truss ($T_{cell} = 5$). Each cell occupies an identical representative (black dotted) volume and comprises two types of struts: inclined (red) and support (gray) struts. These struts are related via a constant $C$, defined as the ratio of the radius of the inclined strut to the radius of the support strut (i.e., $C = r_1/r_2$).

cellular materials (Ⓐ and Ⓑ) and an approximated failure bound (Ⓒ), assuming that available polymeric base materials are isotropic and their post-yield behavior is negligible (inset of Fig. 2a; details of these boundaries described in Supplementary Note 2).

### Stress–strain input curve

Within the full design space formulated above, our generative ML approach takes an arbitrary compressive stress–strain curve, either monotonic or cyclic, as the input. This target curve is constructed via sequentially connecting control points assigned by the user, starting from the origin to linear-elastic limit, followed by local maxima and/or minima, and terminating at the endpoint (the design rules are shown in Fig. 2b; see Supplementary Fig. 4 for an example of the stress–strain curve input process). The first segment of the target curve is a straight line described by two control points at the origin and the linear-elastic limit (i.e., $(\varepsilon_O, \sigma_O)$ and $(\varepsilon_1, \sigma_1)$), representing linear-elastic behavior, and the slope of this straight segment ($E_O$) denotes the elastic modulus of the material under compression. After the linear-elastic segment, the subsequent segments between successive control points ($\varepsilon_i, \sigma_i$) denote the nonlinear behavior of the material, where the maximum number of control point index $i$ is dictated by the initial slope ($E_O$) and given print parameters (see Supplementary Note 3 for details). These control points of the stress–strain curve, along with load type ($T_{load}$) (either monotonic or cyclic), strain energy ($\Delta U$) (area enclosed by the curve), and slope ($E_i$) between two adjacent control points, forms a total of 46 curve features {$\mathbf{X}$} and used as the input to the ML pipeline (details of the curve parameterization presented in Supplementary Note 4).

### Architectural cells for training

We developed a family of cubic symmetric, strut-based architectural unit cells to generate training data sets of our ML approach (Fig. 2c). The cells are represented by design parameters {$\mathbf{Y}$} that describe a lattice architecture, namely, the cell type ($Y_1$ or $T_{cell}$), the characteristic angle ($Y_2$ or $\phi$), and the radius-to-length ratio of the inclined strut ($Y_3$ or $r_1/L_1$) (Fig. 2c). The evolution of $\phi$, together with $r_1/L_1$ tuning, not only changes the relationships among tensile and compressive load-bearing strut members, nodal connectivity, and strut slenderness ratio but also controls the deformation mechanism of the cells, thereby giving rise to distinct stress–strain curves (the rational of the developed cells discussed in Supplementary Note 5; their architectural evolution shown in Supplementary Movie 1; mechanical performance assessment provided in Supplementary Fig. 6; size effects shown in Supplementary Figs. 7, 8). In addition, due to the inherent cubic symmetry of the reported architectural cells, their mechanical behaviors are invariant in three orthogonal directions. This characteristic enables an effective, direct tessellation across different architectures for the creation of compound lattices (i.e., a lattice made of different unit cells) offering enhanced stress–strain curve tunability. Hence, the developed unit cells allow our ML approach to capture diverse stress–strain curve paths while occupying nearly the full range of the design space.

### Training dataset generation

Next, using the developed architectural cells, we generated a training dataset containing design parameters {$\mathbf{Y}$} and their corresponding stress–strain curve features {$\mathbf{X}$} (i.e., {$\mathbf{X}$}-{$\mathbf{Y}$} pairs). We first discretized $\phi$ (or $Y_2$) and $r_1/L_1$ (or $Y_3$) into a number of intervals to create hundreds of basic architectural configurations (training dataset structure listed in Supplementary Table 2). Each configuration was tessellated in three principal directions to create a 3D lattice digital model with the overall dimension of $20 \times 20 \times 20$ mm³ (two unit cells in each orthogonal direction). Three samples were fabricated for each digital model using digital light 3D printing with a brittle polymer (see Methods for its chemical formulation). Stress–strain curves of the as-printed lattice samples were measured by monotonic compression and cyclic compression experiments (measured stress–strain curves illustrated in

Supplementary Fig. 9). The measured stress–strain curve of each architectural configuration was then parameterized into 46 curve feature variables {$X_i, i = 1 \ldots 46$} and paired with the corresponding design parameters[27], leading to 1212 {$\mathbf{X}$}-{$\mathbf{Y}$} pairs in the pristine dataset. As these pairs provide links between the curve features of the experimentally measured curves and the corresponding lattice designs, the training dataset inherently accounts for process variability (e.g., uncertainty and imperfections) stemming from fabrication and experimental measurements. Additionally, as the forward module performs "many-to-one" mapping, we carried out data augmentation on these pairs to account for prediction fluctuation, resulting in 9360 {$\mathbf{X}$}-{$\mathbf{Y}$} pairs in the augmented dataset (details presented in Supplementary Note 6).

### Machine learning model training

For training of our generative ML approach, we first trained the forward module with the augmented dataset. The forward validation module predicts the curve type (via a curve type classifier) and curve features {$\mathbf{X}^P$} (via a curve feature regressor) of given lattice design (details of the forward validation module described in Supplementary Notes 7.1–7.2; details of NN models provided in Supplementary Note 7.4). This forward module effectively acts as a surrogate model that replaces conventional simulations used to evaluate the responses of a design. As compared to the conventional simulation typically requiring hours to compute the mechanical behavior of a 3D lattice design, this forward module takes a few seconds to evaluate the mechanical behavior, which greatly shortens the time span of the entire design process.

Once the forward module was trained, this module was kept frozen (i.e., the weight and bias parameters of all surrogate models were fixed) and was then used to train the inverse prediction module with the pristine training dataset. In the entire training process of the inverse module, the cost function $f_c = (\mathbf{X}^P - \mathbf{X}^T)^2$, which evaluates the difference between the predicted stress–strain curve features {$\mathbf{X}^P$} and the target curve features {$\mathbf{X}^T$} for a given architecture {$\mathbf{Y}$}, was used to optimize the hyperparameters of all the NN models. This strategy prevents any instability during training, which could otherwise cause {$\mathbf{Y}$} to become a meaningless latent space variable (details of the inverse module described in Supplementary Note 7.3). Our ML approach with the optimized hyperparameters showed satisfactory overall prediction accuracy via a typical cross-validation technique (i.e., random 70/30 train/test split). Specifically, each NN model reaches a plateauing loss and eventually features prediction accuracy (~90%) when the training size exceeds about 50% of the dataset with minimal signature of over- and under-fitting, indicating that the training data size is adequate to reasonably satisfy the design goal (training results in Supplementary Fig. 10 for the curve type classifier; Supplementary Fig. 11 for the forward validation module; Supplementary Fig. 14 for the inverse prediction module).

### Inverse design based on various stress–strain curves

Next, using our ML approach, we demonstrated the inverse design of representative stress–strain curve paths of a cellular solid subjected to monotonic and cyclic compression. As illustrated in Fig. 3a, these target curve paths include (i) a linear-elastic section followed by a negative stiffness section, depicting buckling (cases I and V); (ii) a linear-elastic section followed by positive and nearly zero stiffness sections, illustrating strain hardening and plateau regions, respectively (cases II and VI); (iii) a linear-elastic section followed by immediate fracture, characterizing brittle behavior (cases III and VII); and (iv) a linear-elastic section followed by controlled post-buckling, showing a snap-through response (cases IV and VII). These target curve paths (gray solid curves) were significantly different from any curves in the training dataset, which guarantees that our ML approach does not have explicit prior knowledge of these curves (the curve similarity tests

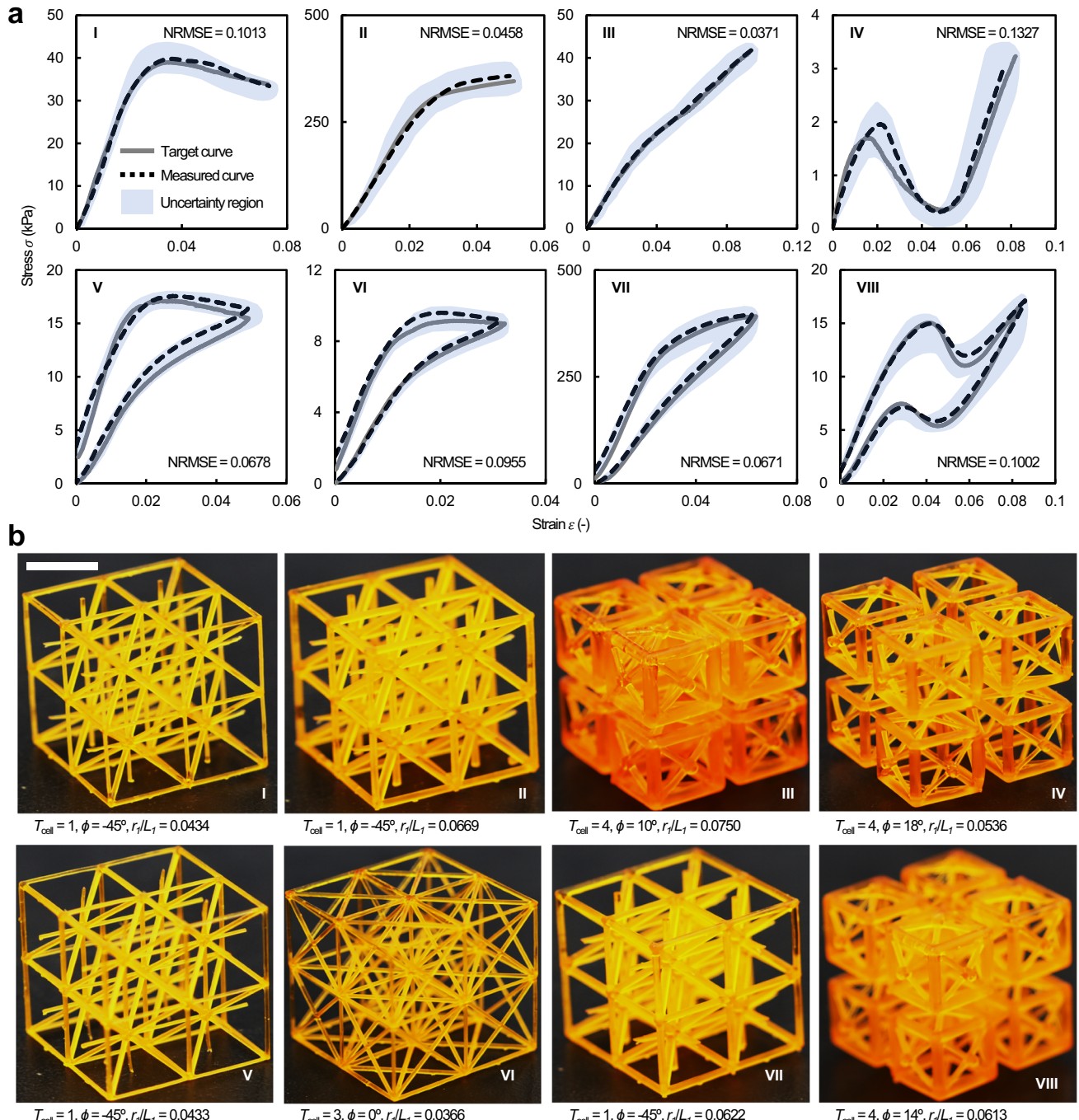

**Fig. 3 | Inverse design based on representative target stress–strain curves and experimental design validation. a** Inverse design based on representative target stress–strain curves describing various compressive responses, spanning linear-elastic behavior followed by either a negative, nearly zero, or positive tangent modulus to a multislope tangent modulus, in response to monotonic (I–IV) and cyclic (V–VIII) uniaxial compression loadings. The gray curves denote the target curves, whereas the black dotted curves represent selected measured curves (from ten measured curves) of the printed samples. The uncertainty region, highlighted by blue shading, covers the distribution of ten experimentally measured curves, illustrating process variability. The normalized root-mean-square error (NRMSE) quantifies the curve similarity between the target and all the measured curves (0: identical curve pair; 1: completely dissimilar curve pair). **b** Photographs of the printed samples inversely designed by the presented ML pipeline in response to each target curve shown in **a**. The ML-predicted design parameters are listed. The scale bar is 10 mm.

between the target curves and all training curves provided in Supplementary Note 7.6; Supplementary Fig. 19d). We fed the curve paths into our ML pipeline to obtain the optimal design parameters, from which ten samples for each curve path were additively manufactured via the same 3D printing apparatus used for our training database generation. Representative printed samples and the predicted optimal design parameters are shown in Fig. 3b.

Results of the inverse design of the representative stress-train curve paths are shown in Fig. 3a. In this figure, the best matching curve (black dotted curve) from ten measured stress–strain curves for each case is compared against the corresponding target curve (gray solid curves), and a blue-shaded uncertainty zone, describing the distribution of the test curves from ten printed samples, represents manufacturing variability. We found excellent similarity between the target

curve and best matching curve for all cases (highlighted by the computed normalized root-mean-square error close to zero), revealing that our method can automatically take into account various manufacturing defects in stereolithography, which vary sample by sample and even strut by strut. This scope is very challenging or impractical to capture with other approaches, such as topology optimization[24].

Our ML approach is also applicable to other polymer-based AM platforms exhibiting larger process variabilities with minimal decreases in reliability. When the process variability ($\eta$), defined as the ratio of the deviation to averaged value of mechanical properties of printed samples, increased by a factor of ~2.6, which makes the printing process used in this study comparable to that of the selective laser sintering process[31–34], the overall prediction accuracy of our ML approach was reduced by ~7%, resulting in an acceptable uncertainty region for the inverse design (details of our process variability study described in Supplementary Note 8; Supplementary Figs. 20–23). While accuracies for recreating materials in response to larger processing errors could be compensated by incorporating larger training data sets, other manufacturing defects, such as anisotropy, porosity, shrinkage, and micro-structural evolution that are unique to metal additive manufacturing not accounted for in the present method (see Discussion section).

## Tailorability of stress–strain curves

Architected materials that meet multiple target properties could be inversely designed via graphically tailoring curve features of a target stress–strain curve, for example by adjusting stiffness, peak stress,

compressibility, and/or nonlinear response (Fig. 4a). To demonstrate tailorability of our design process, we inversely designed an architected shoe midsole by graphically tailoring stress–strain curves measured from a commercial midsole (i.e., baseline curves) for enhanced running performance (detail of baseline curve acquisition described in Supplementary Note 9.1). The midsole was partitioned into four sections upon different levels of loads during heel-toe running[35], and the target stress–strain curve for each section was created by tailoring a baseline curve for the purpose of maximizing running propulsion and cushioning (Fig. 4b, c; the measured baseline curves shown in Supplementary Fig. 24; the design rationale discussed in Supplementary Note 9.2). The tailored midsole consists of a stiff but comfortable toe section, firmer and higher propulsion forefoot section, and stiffer yet energy dissipative heel section. Moreover, the target curves were scaled according to the scaling relationship of the base material (TMPTA) between strain rate and its mechanical properties so that dynamic responses in running scenario can be inversely designed using quasistatic training data (the detailed inverse design of non-quasistatic strain rates illustrated in Supplementary Figs. 25a–d). The as-fabricated midsole sample with optimal design parameters for each section is shown in Fig. 4c (the predicted design parameters in Supplementary Fig. 25e). The results revealed excellent agreement (>90% average prediction accuracy) between the experimentally tested curves and target curves of each tailored section (Fig. 4d; their cyclic responses shown in Supplementary Fig. 25f), indicating that the ML pipeline is capable of creating materials satisfying multiple tailored mechanical responses under different loading conditions.

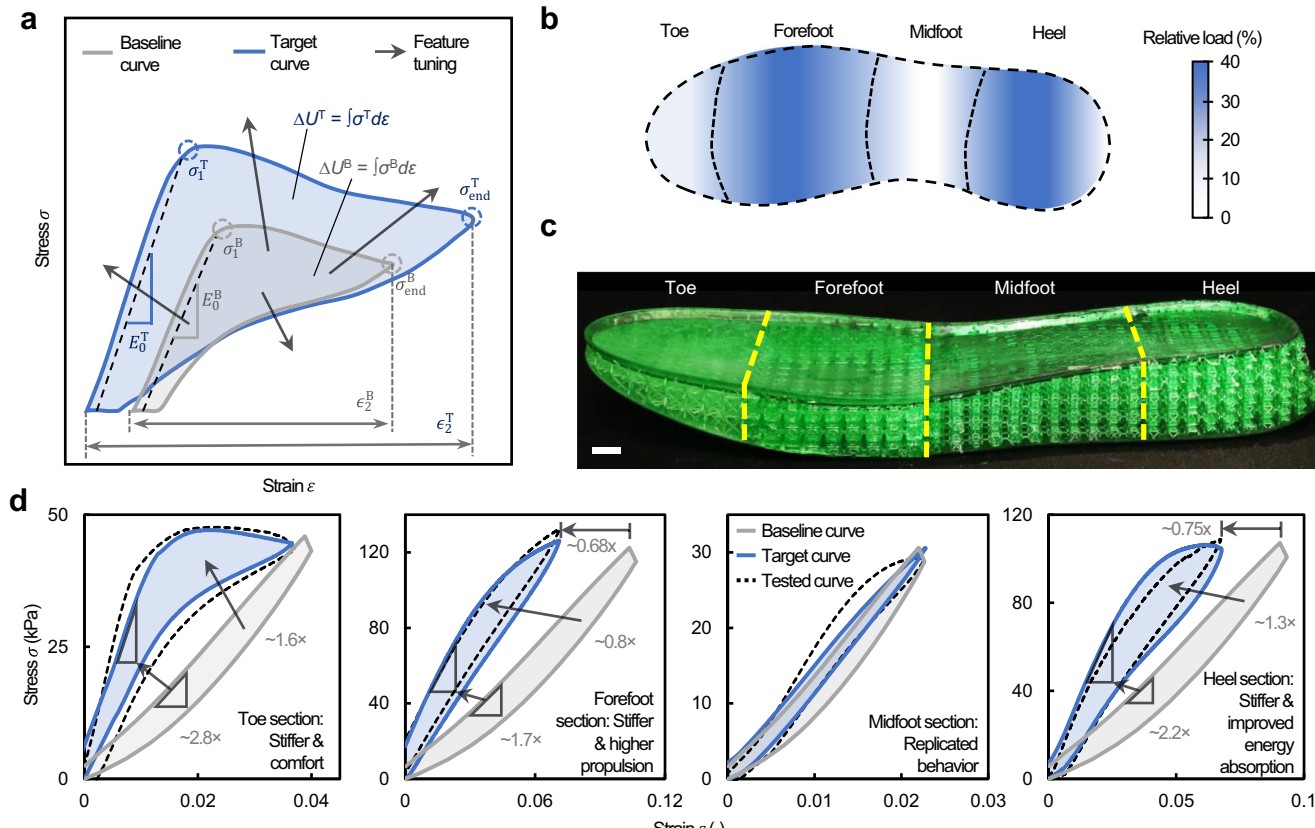

**Fig. 4 | Tailorability of stress–strain curves demonstrated by inverse design of an architected shoe midsole. a** A schematic of the tailoring process to improve the energy absorption behavior. **b** Relative load distribution of the midsole during running[32]. **c** A photograph of the ML-designed architected midsole sample, where each section was designed to exhibit disparate target behaviors. The scale bar is 10 mm. **d** Target and measured stress–strain curves of the architected shoe midsole sample. The baseline response of the commercial midsole (gray curves) for each section was tailored to achieve a specific design target aiming at an improved running performance (blue curves). The tailored curves were then fed into our ML pipeline to obtain optimal designs for each design target, from which the predicted designs were verified via experiments (black dotted curves).

## Enhanced tailorability via compound lattices

The inverse design of architected materials can be further expanded to include advanced curve features that do not exist in natural materials, such as variable tangent modulus, controllable softening/hardening effects, and multiple peaks and valleys. These curve features offer improved crushing behavior and energy absorption performance and could be realized by inversely designing compound lattices (non-uniform lattice comprised of design parameters varying by location) with tailorable mechanical behaviors that go beyond mechanical responses from uniform lattices (lattice materials comprised of identical unit cells throughout the lattice) (see Supplementary Note 10.1; Supplementary Fig. 26). Hence, we created a training dataset containing compound lattices made of a flexible polymeric base material via FE simulations (see Methods). Instead of being represented by uniform design parameters, these compound lattices were described by variation of design parameters (topology gradients) within a confined lattice volume, such as the unit cell type ($G_1$), strut aspect ratio ($G_2$), inclined strut diameter ($G_3$), and cell size ($G_4$) gradients (full description of gradient labels described in Supplementary Note 10.2). Additionally, to adopt the structure of the gradient labels, we employed a sequential integrated strategy for the inverse design process (Supplementary Fig. 27).

To showcase inverse design of advanced curve features discussed earlier, we fed three sets of stress–strain curves into our revised ML pipeline separately, where each case focuses on separately tailoring tangent modulus, first peak stress, and second peak stress (Fig. 5a–c). The corresponding inversely designed 3D digital models describing compound lattices and the spreads of their gradient labels describing the variation of the gradient labels within the lattice in terms of the coefficient of variance are also presented in this figure (the predicted gradient labels in Supplementary Tables 6–8). The ML-predicted results revealed that manipulating pairs of gradient labels independently modulate advanced curve features, including multiple peak stresses and signs of the tangent modulus. This enables fine control of a variety of sectioned stress–strain curves (Fig. 5a–c) not seen with uniform lattices. For example, in the case of tuning tangent modulus (Fig. 5a), we noticed negligible variation in unit cell and strut radius ratio gradients ($G_1$ and $G_2$), indicating the sign of the tangent modulus is mainly controlled by a combination of the inclined strut radius and unit cell size gradients ($G_3$ and $G_4$). Similarly, $G_1$ and $G_2$ together modulated the first peak stress (Fig. 5b). Additionally, in the case of tuning second peak stress (Fig. 5c), significant variation in $G_2$ was observed while the other three gradients almost remained the same, indicating $G_2$ were mainly responsible for the second peak stress manipulation.

Next, we experimentally validated the efficacy of our approach via inversely designing three stress–strain curves which feature different numbers of stress peak and valley events as well as a controlled tangent modulus. These target curves can be found in the insets of Fig. 5d–f, and the primary graphs show the re-created stress–strain curves of the predicted designs (the predicted gradient labels and printed samples in Supplementary Table 8 and Supplementary Fig. 28, respectively; their associated in situ uniaxial compression tests provided in Supplementary Movies 2–4). For the target curve with a nearly zero tangent modulus (the inset of Fig. 5d), the predicted gradient labels indicated minimal variation (Supplementary Table 9), and homogeneous deformation of a designed compound lattice was observed (Supplementary Movie 2). The target curve in the inset of Fig. 5d was then tailored to exhibit a negative tangent modulus after the first peak (the inset of Fig. 5e). The measured mechanical behavior shown in Fig. 5e revealed the localized nonaffine deformation (see Supplementary Fig. 28, Supplementary Movie 3) and corroborated the role of the gradient labels discussed earlier; the first peak stress was dominated by $G_1$ and $G_2$, followed by a subsequent shifting/snapping event with a negative tangent modulus controlled by $G_3$ and $G_4$

(the predicted gradient labels listed in Supplementary Table 9). The target curve shown in the inset of Fig. 5e was further tailored to contain a second peak stress (the inset of Fig. 5f), while keeping all preceding curve features. The predicted gradient labels included a change in $G_2$ (~50%) substantially different from that of the former lattice shown in Fig. 5e, confirming the role of this gradient in peak stress manipulation (the gradient labels listed in Supplementary Table 9; nonaffine deformation shown in Supplementary Fig. 28 and Supplementary Movie 4).

These advanced, inversely designed stress–strain curves featuring successive peak stresses and coordinated collapse mechanisms together with tailored softening effects make the inversely designed compound lattice shown in Fig. 5f an excellent candidate for ML-designed custom padding materials for energy absorption. To test its energy absorption performance, drop tests were conducted on the sample (Supplementary Fig. 29a), showcasing that the measured acceleration and potential energy due to impact were reduced by ~30% and ~25%, respectively, as a result of the compound lattice (Supplementary Figs. 29b, c). In the normalized energy absorption vs. transmitted strength property map (($U/E_s$)/$\bar{p}$ vs. ($\sigma_{tr}/\sigma_{ys}$)/$\bar{p}$), this compound lattice shows energy absorbing performance outperforming that of previously reported lattice materials[36–42] (Supplementary Fig. 29d; the values used in this figure provided in Supplementary Table 10).

## Discussion

This work presents an ML-based rapid inverse design methodology to recreate and tailor mechanical behavior based on stress–strain curves. Our generative ML strategy is capable of mimicking nearly all possible uniaxial compressive stress–strain curves of architected materials, including linear elasticity, strain softening/hardening, tunable tangent modulus, yielding, fracture, tailorable stress peaks and valleys, energy absorption, while accounting for existing 3D process defects, resolutions, and uncertainty. We demonstrated the inverse design of the architected shoe midsole with tunable dynamic performance with spatially tailored sections described by specific stress–strain responses, and also showed enhanced stress–strain curve tailorability by incorporating gradient labels in the ML pipeline, enabling advanced curve features with programmed stepwise energy absorption. Moreover, our ML approach permits optimized structures to be produced with less experimental testing and fast evaluation time. Indeed, a nonlinear stress–strain curve can be analyzed and inversely designed into a 3D digital model from a typical consumer desktop computer within a few seconds with the reported approach, compared to simulations and optimization approaches that would otherwise take days without even taking full account of manufacturing variabilities. Furthermore, while the current work is limited to design compressive behaviors, our ML pipeline could be adapted to inverse design other mechanical responses separately or simultaneously, when accompanied with a family of training data of which each describes a specific loading case (e.g., tensile, compressive, bending, shear, and so on). This is attributed to the fact that the stress–strain curve was adopted as the input, which can describe mechanical behaviors under other types of loading (details in Supplementary Note 12). We also envision that our ML strategy is not limited to mechanical behaviors and can be extended to other complex behavior such as acoustic, magnetic, and electromechanical responses when such responses are expressed in form of a curve similar to the stress–strain curve (e.g., absorbance-frequency, magnetization-magnetic field, polarization-voltage and so on). This work represents progress toward a rapid inverse design and manufacturing methodology that allows for prescribing the full spatial and temporal behaviors of a product that can be printed via a simple desktop computer. It has direct implications for future development of protective wears, automobile and aircraft parts, energy absorbers, and smart materials via simplified design-manufacturing cycle.

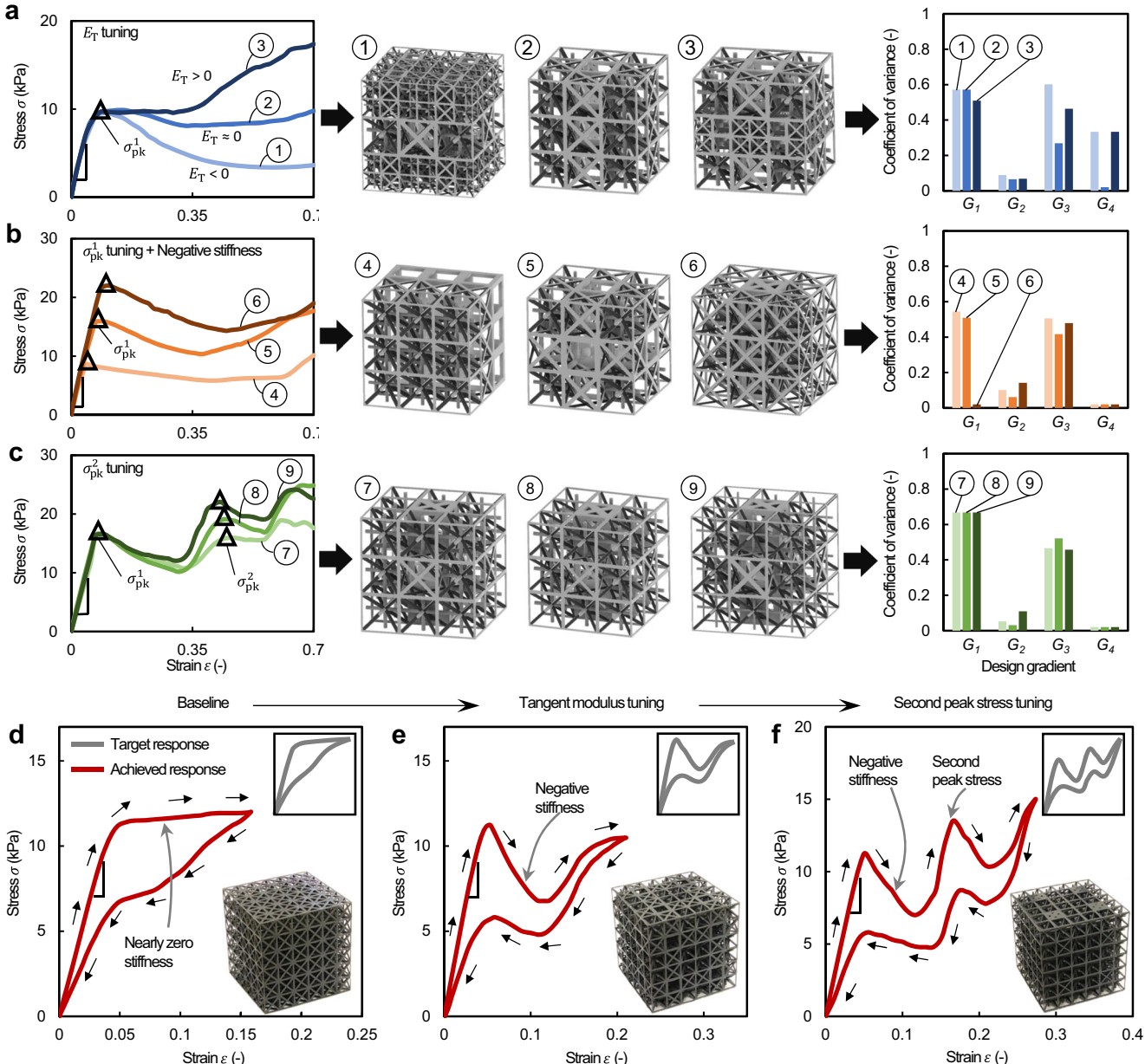

**Fig. 5 | Enhanced stress–strain curve tailorability through compound lattice creations using superposed design gradients. a–c** Three sets of target stress–strain curves, in which the set of curves 1–3 represents the tailorability of the tangent modulus ($E_T$), the set of curves 4–6 represents the tailorability of the first peak stress $\sigma^1_{pk}$ and subsequent negative stiffness, and the set of curves 7–9 represents the tailorability of the second peak stress ($\sigma^2_{pk}$). Insets show 3D digital models representing compound lattices inversely designed by the ML-predicted design gradients ($G_1$: unit cell gradient, $G_2$: strut radius ratio gradient, $G_3$: inclined strut radius gradient, $G_4$: unit cell size gradient) for the target curves, and their coefficients of variance characterizing the spread of the corresponding design gradients are also shown. **d–f** Experimental demonstration of tailored stress–strain hysteresis loops displaying deformations at different strains. The stress–strain hysteresis loop in **d** was tailored to exhibit negative stiffness in **e** and further tailored to exhibit multiple stress peaks in **f**. The target curves and photographs of the as-fabricated, ML-designed compound lattices are shown in the insets.

We demonstrated that experimental data for ML automatically learns processing incurred errors; as a result, the re-created metamaterials replicate the prescribed mechanical behaviors with high accuracy after printing. We note that the experiments in this work were carried out via a mass-consumer based, desktop printing technique using projection micro-stereolithography with polymeric materials, which comes with inherently higher fabrication accuracy compared to other additive manufacturing processes (e.g., laser sintering for metal printing). While one can project that method such as using a higher amount of training data or employing transfer learning to other printing methods (e.g., metal printing) to mitigate potentially larger errors from manufacturing, it remains a future topic of study how variations of base material property that are inherent to metal processing, such as sintering, anisotropy and micro-structures of metals, will influence the fidelity of the replication process[43]. This could potentially be mitigated by adding process parameters as the input, such as energy density and powder size, which could help our ML pipeline account for the variabilities brought by stochastic interaction between mass and energy in the process. Additionally, the current study didn't add features including property evolution and anisotropy effect resulting from material processing, where such effects are minimal in the process employed here but could have larger variations in other processing methods.

## Methods

### Sample fabrication

All lattices presented in this work were created as 3D digital models using commercial computer-aided design software (Autodesk Inventor 2022). Two distinct base materials were used for fabrication, where each base material was assigned to a unique dataset. A brittle base material, denoted by TMPTA, was a photosensitive resin consisting of trimethylolpropane triacrylate (Sigma–Aldrich Inc., St. Louis, MO) with 0.0125 wt% photoabsorber and 2 wt% phenylbis(2,4,6-trimethylbenzoyl) phosphine oxide photoinitiator (Sigma–Aldrich Inc., St. Louis, MO), whereas a flexible base material was a commercial resin (Flexible 80 A, Formlabs Inc., Somerville, MA) and used as received. A digital light 3D printer (Anycubic Photon S, Anycubic Inc., Shenzhen, China) was used to fabricate brittle samples using TMPTA, and a customized projection stereolithography system was built and used for printing flexible samples. At least three samples for each model were tested during the validation process. After fabrication, all samples were cleaned with ethanol and dried in a dark environment for 24 hours.

### Experimental testing

All quasistatic compression tests were performed by using the Instron 5944 universal testing machine (Instron Corporation, Norwood, MA). The printed lattice samples were compressed between the stationary and moving steel plates. The loads were measured by the Instron load cell with a load capacity of 2000 N (serial no.: 150821), and the displacements were measured by the built-in encoder associated with the crosshead movement. For the monotonic compression test, stress–strain curves of the samples were recorded up to the onset of the first appearance of failure. For the cyclic compression test, hysteresis loops with three different strain levels were recorded while ensuring elastic recovery of the samples. The strain rate for all tests was set as $10^{-3} s^{-1}$. Stress was computed as the measured load divided by the effective area ($L_{lattice}$)$^2$, and strain was calculated as the displacement divided by $L_{lattice}$, where $L_{lattice}$ refers to the side length of the sample.

Drop tests were performed by dropping a dead weight (mass $m$ of 500 g) onto the test samples from different heights, $h$. The sample size used in the drop tests was nominally $60 \times 60 \times 60$ mm$^3$ in volume. The drop height was between 50 and 150 mm to measure energy absorption while preventing damage to the test samples. The impact force was recorded by a force transducer fixed directly underneath the bottom of a flat, rigid steel plate. The transmitted force was measured in a similar manner but with the sample fixed on the top of the rigid plate. Both measurements were taken at a sampling rate of 100 Hz using the Instron data acquisition hardware. Impact acceleration was calculated using Newton's second law, $a = F/m$, where $F$ is the measured force and $m$ is the mass of the dead weight. The potential energy was computed as $P = Fh$.

### ML model setups and evaluation

The ML models in the inverse prediction module and forward validation module were implemented using common NN models (a generative model and surrogate model, respectively) on Python 3.7 for the general applicability of our ML approach. During training, hyperparameter settings for all models were optimized by a stochastic gradient descent optimizer (Supplementary Note 7). Prediction accuracy of the ML pipeline was evaluated by adopting a typical 10-fold cross-validation technique with the optimized hyperparameters and all training data instances, which is equivalent to a training/testing split ratio of 70:30 with interchangeable switching of the training and testing sets (results of the 10-fold cross-validation in Supplementary Note 7).

### Sequential integration strategy for compound lattice prediction

The sequential integration strategy was adopted for inverse design of compound lattices (i.e., lattices with dissimilar unit cells), where a subset of the previously predicted design gradients was utilized as a part of the input in subsequent prediction stages. Specifically, in the first step of training of the ML model comprising two classifiers and two regressors (Supplementary Fig. 27), the unit cell gradient ($G_1$) was estimated via a classifier for curve features {$\mathbf{X}$} parameterized from a target stress–strain curve. Then, a compound descriptor combining {$\mathbf{X}$} and the predicted $G_1$ were used to determine the strut radius ratio gradient ($G_2$) via regression. A subsequent regression task was performed to estimate the inclined strut radius gradient ($G_3$) using another compound descriptor combining {$\mathbf{X}$} and the predicted $G_1$ and $G_2$. As the last step, the unit cell size gradient ($G_4$) was classified by using a compound vector composed of {$\mathbf{X}$} and the predicted $G_1$, $G_2$, and $G_3$. This process can be summarized as follows: (i) $C_1(\mathbf{X}) \rightarrow G_1$; (ii) $R_1(\mathbf{X}, G_1) \rightarrow G_2$; (iii) $R_2(\mathbf{X}, G_1, G_2) \rightarrow G_3$; and (iv) $C_2(\mathbf{X}, G_1, G_2, G_3) \rightarrow G_4$, where $C$ and $R$ denote classification and regression, respectively.

### Finite element simulations

FE simulations of the stress–strain curves of lattices were conducted using ABAQUS 6.14. Lattices at low- and mid-relative densities ($\bar{p} \leq 0.15$) were discretized using 2-node linear Timoshenko beam elements (B31 of ABAQUS), whereas second-order tetrahedral elements (C3D10M of ABAQUS) were used to discretize lattices at higher relative densities ($\bar{p} \geq 0.15$) (Supplementary Fig. 30). This is attributed to the fact that the beam element is not capable of accurately describing the warping of cross-section of stubby struts whereas 3D stress element is able to precisely capture the deformation of the lattices with high relative densities[44]. The mesh seed sizes for both the beam and solid models were determined from our mesh sensitivity study; each strut modeled by between 15 and 30 elements depending on its length. Each lattice was compressed between a fixed and a moving, flat, rigid surface discretized by rigid bilinear quadrilateral elements (R3D4 of ABAQUS). The explicit solver was used while keeping the kinetic energy less than 1% of the total internal energy to ensure a quasistatic loading condition. A 10% percentage of mode shapes (eigenmodes) of the lattice were applied to the simulation to prevent bifurcation issue at the buckling point, which might cause non-convergence solution in the numerical analysis. Contact effects were modeled using a hard contact behavior for the normal direction and finite sliding in the tangential direction with a coefficient of friction of 0.8. For the constituent material models, TMPTA, used to create periodic lattices, was modeled as an elastic–plastic material (a short plastic region after linear-elastic region representing brittle fracture) with isotropic hardening, Poisson's ratio $v_s$ of 0.3, and a fracture strain of 0.044. Formlabs Flexible, used to create compound lattices, was modeled as a linear-elastic material with Poisson's ratio $v_s$ of 0.48 and a fracture strain of 0.424. These material models were verified by the measured tensile response of the dogbone test samples printed with such base materials (Supplementary Fig. 31).

## Data availability

The data supporting the findings of this study are available in the Supplementary Information. The pristine and augmented training data sets used in this work are available at https://doi.org/10.5281/zenodo.8210037.

## Code availability

The source code used in this work is available at https://doi.org/10.5281/zenodo.8210037. The setup and hyperparameters can be found in Supplementary Note 7.

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

## Acknowledgements

The authors gratefully acknowledge the Office of Naval Research (Grant No. N00014-20-1-2504:P00001), National Science Foundation DMREF grant (Grant No. 2119643), and the Air Force Office of Scientific Research (Grant No. FA9550-18-1-0299). The research reported in this publication was partially supported by the Department of Defense (Grant No. N00014-19-1-2728).

## Author contributions

X.Z. conceived the research. X.Z., Z.K. V.D., and M.B. guided and supervised the research. C.H. developed the framework for stress–strain curve inverse design. C.H. and D.Y. designed a family of

architectural cells with cubic symmetry. C.H. and M.K. developed the feature extraction technique. C.H., C.L., Z.K., D.Y., H.L., and M.B. implemented the machine learning framework. C.H. and H.L. performed training on the ML models and prediction of design parameters. C.H. and Z.X. designed the experiments. Z.X., C.H., D.Y., and D.E. fabricated the samples, measured the mechanical properties, and performed the data analysis. Z.X. and D.Y. developed the gray-mask technique. C.H. and D.Y. performed the analytical, numerical calculations, and finite element simulations for training and verification. C.H. and D.Y. implemented the design strategy of the compound lattice with gradient labels. C.H. wrote the initial manuscript with help from co-authors. C.H., X.Z., D.Y., D.E., and Z.X. edited the manuscript with input from all authors. All authors participated in the discussion and interpretation of the data.

## Competing interests

The Regents of the University of California, a California Corporation has filed a U.S. provisional application (serial no. 63/456200) for the presented machine learning-based rapid inverse design methodology (pending). Inventors include X.Z., C.H., D.Y., and M.B. The remaining authors, Z.X., C.L., H.L., D.E., M.K., V.D., and Z.K. declare no financial and non-financial competing interests in the subject matter or materials discussed in this article.
