## [Peer Review File · Nature Communications]

Rapid Inverse Design of Metamaterials based on Prescribed Mechanical Behavior through Machine LearningEditorial Note: This manuscript has been previously reviewed at another journal that is not operating a transparent peer review scheme. This document only contains reviewer comments and rebuttal letters for versions considered at *Nature Communications*.

REVIEWER COMMENTS

Reviewer #1 (Remarks to the Author):

The authors have thoroughly revised their manuscript and added a significant body of new and helpful information. Especially the provided details on the curve parameterization and on the (newly revised) ML framework are appreciated. The quality of the manuscript has hence improved considerably, yet the following points should still be considered:

1. The manuscript, especially in the title, still overstates the generality of the findings; e.g., the model only predicts a 1D uniaxial compressive response (not the general "mechanical behavior"), it only applies to 3D-printed polymers (as also pointed out by reviewer 2), etc.
2. The newly revised ML approach, consisting of a forward surrogate NN and an inverse NN, is not novel nor original but was in fact introduced in reference [19] of the manuscript (Kumar et al.). [19] focused on elastic stiffness but otherwise used the same inverse design setup of two sequentially trained NNs for forward and inverse maps to overcome exactly the non-uniqueness addressed here. It would be warranted to properly credit in the manuscript and SI where this approach is adopted from.
3. The training set seems tiny compared to comparable inverse design approaches in the literature. Could the authors provide a reasonable explanation for why such a small data set is sufficient for training?

Detailed response to Reviewer 1

We thank the referee for the time in conducting the review. We have addressed the reviewer's comments one-by-one below. Within the manuscript and Supplementary Information itself, the text where key revisions have been made is marked in red.

Reviewer #1 (Remarks to the Author):

1. The manuscript, especially in the title, still overstates the generality of the findings; e.g., the model only predicts a 1D uniaxial compressive response (not the general "mechanical behavior"), it only applies to 3D-printed polymers (as also pointed out by reviewer 2), etc.

Response: We agree with the reviewer and, here, we comment on how the current model can be extended to other loading cases. We envision that the presented ML pipeline can be readily extended to inverse-design other mechanical behaviors, since the stress-strain curves used herein as input could represent other types of loading such as tension, shear, torsion, and bending, or different strain rate conditions.

In detail, the ML pipeline can be applied to inverse-design of metamaterials that replicates other mechanical responses individually (Fig. R1). This can be achieved by adopting a collection of training datasets (i.e., training databank), wherein each of which contains a specific loading case (e.g., tensile, compressive, bending, shear) linked to their corresponding metamaterial topology. The sample size for each dataset is expected to be similar to the one used in the manuscript. A user can then pass a desired response (in form of stress-strain curve) along with a newly introduced input variable (Π), classifying a loading type, onto the ML pipeline. For instance, when Π is 1, a tensile dataset is selected from the databank to train the ML pipeline. Although, during training, the hyperparameters of the neural network models need to be tuned to specifically describe the nature/complexity of the mapping relationship of interest (i.e., curve features-to-metamaterial topology) so as to maximize the validation accuracy, we expect the structure of the present ML pipeline and the training process to remain the same as described in the present manuscript. Once training is done, the target stress-strain curve of interest can then be inverse-designed following the same process as that introduced herein for the compressive loading case.

Beyond individual loading cases, the ML pipeline can be further modified to inverse-design metamaterials that satisfy multiple mechanical behavior inputs simultaneously (i.e., "multi-objective" inverse design). **Fig. R2** illustrates an example of a modified ML pipeline that can inverse-design some metamaterials that simultaneously feature targeted tensile and shear stress-strain curves. The modified ML pipeline is structured so that the inverse design module now involves with two surrogate forward validation modules where each of the forward modules aims at predicting one of these two stress-strain curves associated with both of the loading cases. For training, datasets from the databank corresponding to loading types of the target curves are first selected by utilizing the loading type identifier input (Π) described by an array (e.g., [1,4]). These datasets are then used to create a joint training set, which maps a metamaterial topology to multiple responses (e.g., tensile and shear responses). During training, the hyperparameters will need to be adjusted to achieve satisfactory validation accuracy. Once

the target curves are fed into the modified ML pipeline, the inverse design module predicts five sets of design candidates, which will be passed to the forward validation module to estimate the response of the design candidates, as described in the manuscript. For each design candidate, the total normalized root-mean-square error (NRMSE) value (calculated as the weighted sum of the NRMSE associated with each loading case) can be used to select the optimal design. Indeed, a detailed future study is required to realize how to (1) combine the joint dataset, (2) determine the training data size required for each loading type, and (3) identify the weight assigned to each loading case to ensure that none of the mechanical behaviors of interest dominates the others in the total cost function during the optimal design selection. Furthermore, similar to the design rules used for the plottable compressive stress-strain curve input described in the manuscript (as well as **Supplementary Information section 3**), additional design rules may be necessary to simultaneously inverse-design other mechanical responses.

Based on the reviewer's recommendations, we emphasized the uniaxial compression used in the present manuscript and commented on the possibility of incorporating other mechanical loading scenarios in the **Discussion section** and **Supplementary Information section 12**. We also agree with the reviewer that our ML pipeline is within the scope of additive manufacturing of polymer materials. While the inverse design method can be extended to other families of materials, other materials and process (for example, incorporating metal) will require a future study to take into account the effect of micro-structure changes and defects associated with those processes. To incorporate the reviewer's concerns, we have toned down the generality of our findings in the revised manuscript and have thoroughly revised the discussion section to discuss the limitations and future possibilities. The revised manuscript now reads:

- “Furthermore, while the current work is limited to design compressive behaviors, our ML pipeline could be adapted to inverse-design other mechanical responses separately or simultaneously, when accompanied with a family of training data of which each describes a specific loading case (e.g., tensile, compressive, bending, shear, and so on)....” see discussion section.
- "Specifically, a user sketches a target uniaxial compressive stress–strain curve described by a set of curve features as the input into our generative machine learning pipeline..." on summary paragraph page 2
- "...enables the rapid creation of nearly all possible uniaxial compressive stress–strain curve cases subjected to compression..." in summary paragraph on page 2
- “Our generative ML strategy is capable of mimicking nearly all possible uniaxial compressive stress–strain curves of architected materials...” on page 17
- “...a rapid inverse design methodology leveraging generative machine learning and additive manufacturing with polymeric materials, which...their mechanical behaviors” in summary paragraph on page 2
- “To address the aforementioned issues, we present a rapid inverse design methodology that...via ML and polymer AM while incorporating process variabilities” on page 4
- “Our ML approach is also applicable to other polymer-based AM platforms exhibiting larger process variabilities with minimal decreases in reliability” on page 12
- “...using projection micro-stereolithography with polymeric materials, which comes with inherently higher fabrication accuracy compared to other additive manufacturing processes (e.g., laser sintering for metal printing)” on page 18

Fig. R1. Illustration of the ML pipeline for the inverse design of other single mechanical responses. The input consists of a target curve and a new input variable classifying a loading type, denoted by Π . This variable guides the ML pipeline to select a relevant dataset from the training databank. The inverse prediction and forward validation modules are then trained by the selected data, wherein the hyperparameters of all the neural networks need to be tuned for each specific mechanical response. Once both modules are trained, the rest of the prediction process for the target curve remains the same as described in the manuscript.

Fig. R2. Example of the modified ML pipeline for the simultaneous inverse design of multiple mechanical responses. The target stress-strain curves associated with the loading conditions of interest are parameterized into two curve feature sets $\{X^{Ti}\}$ ($i = 1, 4$, denoting the loading type identifier), which is then combined and used as input for the ML framework. The inverse prediction module predicts five sets of design candidates $\{Y^k\}$ (described by the cell type (T_{cell}), characteristic angle (ϕ), and radius-to-length ratio (r_1/L_1)), where k ranges from 1 to 5. These design candidates are then passed to the forward validation module, where each inverse neural network model is linked with two surrogate models (each contains a curve type classifier and a curve feature regressor), to estimate the response $\{X_k^{Pi}\}$ of the design candidates. For each design candidate, the total NRMSE value (calculated as the weighted sum of the NRMSE associated with each loading case) is used to select the optimal design. The weight assigned to each loading case would need to be adjusted to ensure that none of the mechanical behaviors of interest dominates the others in the total cost function.

2. The newly revised ML approach, consisting of a forward surrogate NN and an inverse NN, is not novel nor original but was in fact introduced in reference [19] of the manuscript (Kumar et al.). [19] focused on elastic stiffness but otherwise used the same inverse design setup of two

sequentially trained NNs for forward and inverse maps to overcome exactly the non-uniqueness addressed here. It would be warranted to properly credit in the manuscript and SI where this approach is adopted from.

As pointed by the reviewer, we have cited the following references. The sentence ending “...solves the nonunique response-to-design mapping challenge in inverse design.” now cites:

- Liu, D., Tan, Y., Khoram, E., & Yu, Z. (2018). Training deep neural networks for the inverse design of nanophotonic structures. *ACS Photonics*, 5(4), 1365-1369.
- Kumar, S., Tan, S., Zheng, L., & Kochmann, D. M. (2020). Inverse-designed spinoid metamaterials. *npj Computational Materials*, 6(1), 73.

3. The training set seems tiny compared to comparable inverse design approaches in the literature. Could the authors provide a reasonable explanation for why such a small data set is sufficient for training?

Response: We appreciate the reviewer’s comment. When implementing our generative ML pipeline, we noticed that the original raw dataset was not large enough to train the forward “many-to-one” mapping model with satisfactory prediction accuracy. Thus, we adopted a data augmentation technique to enrich the original raw dataset so as to increase the number of datapoints available to train the forward model. This was achieved by adopting a standard SMOTE oversampling approach^{1,2}, which has shown some success in a large variety of applications from different domains. As described in the manuscript (**Training data generation** section and **Supplementary Information section 6**), after the data augmentation, the total size of data points increases to more than ten thousand input-output pairs (specifically, 1,212 pristine pairs and about 9,360 augmented pairs), which shows a comparable order of magnitude as compared to previous studies^{3,4}. Notably, this treatment was found to be effective as it resulted in a significant improvement in the overall prediction accuracy—since the final loss of the model decreases by ~50% after data augmentation.

Additionally, we verified that the size of dataset used in this work is sufficient to achieve satisfactory overall prediction accuracy and hence allow the ML pipeline to inverse-design an architecture that matches each given target stress-strain curve. Specifically, we investigated the influence of the training size by constructing a learning curve (wherein the test set accuracy is tracked as a function of the size of the training set) for each of the individual neural network models used in our ML pipeline (see **Fig. R3** below, and, in the Supplementary Information, **Fig. S10** for the curve type classifier, **Fig. S11** for the forward validation module, and **Fig. S14** for the inverse prediction module). It is evident that each of the models exhibits a plateauing loss and eventually features satisfactory prediction accuracy when the training size exceeds about 50% of the entire dataset. Importantly, these learning curves do not exhibit any signatures of over- or under-fitting (i.e., they all feature comparable training and test set loss, and final loss less than ~0.1 at most). Note that the forward module and curve type classifiers utilize the augmented 9,360 instances whereas the inverse module leverages the original raw 1,212 instances. These findings also show that all models exhibit a boost in prediction accuracy upon increasing training set size, which demonstrates the value of our data augmentation approach and confirms that the current dataset is large enough to ensure the accuracy of the present inverse design ML pipeline.

Our response has been incorporated into the revised manuscript, which begins on page 10 with “Our ML approach with the optimized hyperparameters...for the inverse design module)”.

Fig. R3: Learning curves of the neural network models used in the presented ML pipeline. **a**, Learning curves showing the misclassification fraction for the curve type classifier. **b**, Final loss for the forward validation module. **c**, Final loss for the inverse prediction module (for both the training and test sets) as a function of the size of the training set.

References

- 1 Chawla, N. V., Bowyer, K. W., Hall, L. O. & Kegelmeyer, W. P. SMOTE: Synthetic Minority Over-sampling Technique. *Journal of Artificial Intelligence Research* **16**, 321-357 (2002). <https://doi.org:10.1613/jair.953>
- 2 Fernandez, A., Garcia, S., Herrera, F. & Chawla, N. V. SMOTE for Learning from Imbalanced Data: Progress and Challenges, Marking the 15-year Anniversary. *Journal of Artificial Intelligence Research* **61**, 863-905 (2018). <https://doi.org:10.1613/jair.1.11192>
- 3 Liu, D., Tan, Y., Khoram, E. & Yu, Z. Training Deep Neural Networks for the Inverse Design of Nanophotonic Structures. *ACS Photonics* **5**, 1365-1369 (2018). <https://doi.org:10.1021/acsphotonics.7b01377>
- 4 Kumar, S., Tan, S., Zheng, L. & Kochmann, D. M. Inverse-designed spinodoid metamaterials. *npj Computational Materials* **6** (2020). <https://doi.org:10.1038/s41524-020-0341-6>

REVIEWERS' COMMENTS

Reviewer #1 (Remarks to the Author):

The authors have responded to all points raised by this reviewer and have modified the manuscript accordingly. Especially the newly added information on extensions of the work is appreciated and has improved the originality and strength of the manuscript.